# Last Minute at the GermEval-2025 LLMs4Subjects Task: Few-Shot Contrastive Learning for Multilingual Multi-Label Classification

**Parisa Shirali**[*]**, Zahra Sarlak**[*]**, Ebrahim Ansari**

Institute for Advanced Studies in Basic Sciences (IASBS), Zanjan, Iran

p.shirali@iasbs.ac.ir, z.sarlak@iasbs.ac.ir, ansari@iasbs.ac.ir

## Abstract

This paper addresses the challenges and recent advancements in automated domain classification for digital libraries, with a focus on integrating multilingual text embeddings to improve the classification of bilingual technical records. We employed the FastFit package to fine-tune a model capable of performing multi-label classification on the TIBKAT dataset, which presents a highly skewed distribution of subject categories. To enhance label interpretability, we generated detailed class descriptions using GPT-4-o Mini. Our method combines few-shot learning with a modified score calculation to ensure comparability across labels, enabling effective and scalable classification without extensive fine-tuning.

## 1 Introduction

Digital libraries are changing the ways we access and share scholarly and cultural resources by bringing together various information formats. As technology advances, our strategies for utilizing these evolving information technologies must adapt to support lifelong learning. This shift underscores the increasing need to explore automated methods for subject indexing and classification, which are vital for managing the growing volume of digital documents (Greenstein, 2000).

Models trained on extensive datasets can effectively capture complex interdisciplinary subjects, provide cohesive text, and produce useful annotations across various natural language processing (NLP) contexts (Dale, 2021). While these models promise speed and scalability, questions regarding their ability to produce high-quality annotations still remain. For example, what limitations do LLMs encounter in generating human-like annotations? Can they match or exceed the quality of annotations created by humans in bilingual subject tagging? In what manner does training data

influence their ability to accurately capture intricate sentiment relationships in complex topics without inaccuracies or generalizations? These queries are central to the LLMs4Subject task (D'souza et al., 2025).

This challenge invited researchers to develop LLM-based systems that improve domain classification and semantic indexing of bilingual technical records at the TIB (Leibniz Information Centre for Science and Technology). The second LLMs4Subjects task highlights the importance of efficiency in LLMs while focusing on better model performance, reduced energy consumption, and faster inference speed.

In sub-task 1, our aim was to create an efficient model by using few-shot fine-tuning combined with synthetically generated data from readily available instruction-tuned LLMs. The model categorizes human-readable records into one or more of the 28 predefined subject domains from TIB's LinSearch Subject Classification System. Our findings show that few-shot fine-tuning with synthetic data can achieve strong performance without the need for extensive task-specific fine-tuning, providing insights into its potential and limitations for domain-specific classification tasks.

The rest of this article is organized as follows: Section 2 (Related Works) reviews key principles and existing methods in automated subject tagging. Section 3 (Methodology) outlines our approach, Section 4 (Results) presents our findings, and Section 5 (Future Work) discusses implications and future research directions.

## 2 Related Works

Automatic classification of scholarly literature is a critical yet challenging task, particularly in multi-label settings. Prior work has approached this problem using either supervised or unsupervised methods (Gialitsis et al., 2022); (Sadat and Caragea,

---

[*]These authors contributed equally to this work

2022); (Liu et al., 2021); (Toney and Dunham, 2022); (Salatino et al., 2019); (Mustafa et al., 2021); (Kandimalla et al., 2020). However, Generative LLMs like GPT and BERT offer more powerful and flexible few-shot classification capabilities, reducing reliance on large labeled data.

Niraula et al. (Niraula et al., 2024) confirm this ability and show that fine-tuned LLM models trained on a relatively small amount of human-labeled data outperform baseline models. A particularly efficient approach is SetFit (Tunstall et al., 2022) which finetunes a sentence transformer in a contrastive manner without the need for prompt engineering or full fine-tuning. FusionSent (Schopf et al., 2024) is also a few-shot multi-label classification method that fine-tunes dual sentence embedding models from the same PLM checkpoint. One model optimizes intra-class similarity using cosine similarity loss, while the other focuses on aligning instances with labels using contrastive loss (Hadsell et al., 2006). The models are then combined using Spherical Linear Interpolation (SLERP) (Shoemake, 1985) to address limitations such as catastrophic forgetting (Biesialska et al., 2020) then frozen and employed as a logistic regression classification head (Cox, 1958). However, their approach introduces a trade-off in higher computational costs than lightweight alternatives like SetFit.

In this study, we adapt the FastFit package[1], which offers notable improvements in both training speed and classification accuracy compared to existing few-shot learning methods (Yehudai and Bendel, 2024). To improve robustness, FastFit augments the training batch by including the class name as an additional example and repeating each text $r$ times (where $r$ is a positive integer) while applying dropout, a minimal representation-level augmentation inspired by (Gao et al., 2022).

## 3 Methodology

For Subtask 1—Multi-Domain Classification of Library Records—we used the FastFit package (Yehudai and Bendel, 2024) to fine-tune an open-source embedding model.

The TIBKAT collection shows a significantly skewed distribution across its subject domain categories, as represented in Figure 1. If we train a model on this dataset, it will likely become heavily biased toward the majority classes. To mitigate it and regarding FastFit's ability in few-shot learning,

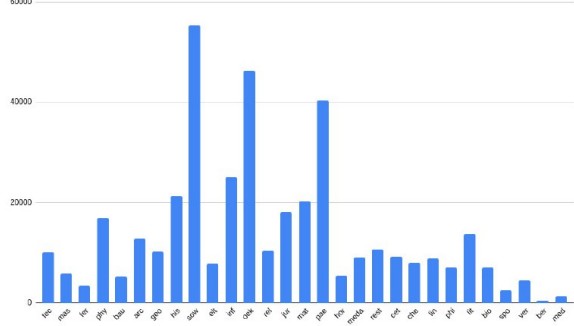

Figure 1: Distribution of domain categories within the dataset. The distribution is notably skewed, with some categories significantly outnumbering others, indicating a potential bias that may impact classification performance.

we randomly selected a fixed number of 100 samples from each subject category. FastFit encodes labels as positive pairs with their corresponding inputs, mapping them into a shared embedding space. It uses in-batch contrastive learning to bring the documents closer to their labels while simultaneously pushing all other texts apart based on their token-level similarity metric. This approach has a more pronounced effect for smaller models and less data, which makes the choice of 100 examples per subject category, a balanced choice. It is sufficient to capture intra-class variation while small enough to equalize representation across categories. Furthermore, its augmentation strategy, which includes class names and repeated examples, further compensates for reduced data volume. However, the domain labels in the TIBKAT dataset are single-word labels, which are often too general to capture the necessary semantic nuances in the embedding vectors. To enhance label clarity, we leveraged GPT-4-o Mini (see Appendix B for prompt) to generate detailed class descriptions for each category, providing one or two sentences for better context[2]. On average, each output contained approximately ±104 tokens, as measured with the OpenAI Playground[3].

Since our classification problem is a multi-label problem, we adjusted how scores are calculated in FastFit. In the original version (Yehudai and Bendel, 2024), FastFit sums the maximum token-level similarity between the input and the label description, which can result in different score scales depending on the number of tokens. This is not an

---

[1] https://github.com/IBM/fastfit

[2] The generated descriptions is available *here*.
[3] https://platform.openai.com/tokenizer

issue for single-label problems, where we simply choose the label with the highest score. However, when selecting multiple labels based on a threshold, we need all scores to be on the same scale. To address this, we normalized the FastFit scores.

We chose the Multilingual-E5 large-instruct[4] (Wang et al., 2024) model to fine-tune, which has performed well on the MTEB (Muennighoff et al., 2023) German benchmark (Wehrli et al., 2024) at the time. The Multilingual E5 Text Embeddings series is recognized for generating high-quality and dense embeddings and exhibits strong multilingual capabilities for encoding and understanding German and English texts.

For each document, we extracted the title, abstract, and classification label from TIBKAT records and saved them in the following format:

- **Query:** The combined title and abstract text, prefixed with "retrieval.query" as the model's text column.

- **Label:** The document's classification label, mapped to its corresponding descriptor (from GPT-4 definitions) and prefixed with "retrieval.passage" as the model label column.

Each query was then organized into separate pairs with its labels as follows:

$(query_1, label_1),$
$(query_1, label_2),$
...
$(query_i, label_k)$

And finally, we fine-tuned the model using the table 1 parameters:

| Training Prameters | Value |
|---|---|
| Optimizer | AdamW |
| Training Epochs | 4 |
| Number of Repeats | 3 |
| Training Batch | 16 |
| Evaluation Batch | 16 |
| Learning Rate | 1.5e-04 |
| Weight Decay | 0.02 |
| Mask Probability | 0.1 |
| Gradient Accumulation Steps | 2 |

Table 1: Training parameters utilized for fine-tuning the model.

[4]https://huggingface.co/intfloat/multilingual-e5-large-instruct

After fine-tuning the model, we calculated the normalized similarity score for each test sample by comparing the concatenated title and abstract with each label description. Labels were assigned based on a selection threshold that was determined to yield the best results in comparison with a limited collection of other thresholds in a range during the evaluation phase, which was set at 0.90.

To track the environmental impact of our training, we used the CodeCarbon package[5]. The Code-Carbon package, monitors CPU, GPU, and RAM consumption and converts the energy consumed into $CO_2$ emissions. As can be seen in table 2

| Training Metrics | Value |
|---|---|
| Energy consumed for RAM | 1.242 Wh |
| RAM Power | 10.0 W |
| Delta energy consumed for CPU with constant | 0.143 Wh |
| CPU power | 42.5 W |
| Energy consumed for All CPU | 5.279 Wh |
| Energy consumed for all GPUs | 8.344 Wh |
| Total GPU Power | 67.95 W |
| Electricity used since the beginning | 14.865 Wh |

Table 2: Training metrics detailing energy consumption and performance statistics for 4 epochs and 2800 training samples taking 8 minutes and 48 seconds.

## 4 Results

The GermEval team evaluated our subject tagging system. Table 3 shows the overall results, including Macro and Micro precision, recall, and F1 scores and Table 4 shows the overall results grouped by language.

Detailed results for different record types and languages can be found in appendix A Table 5.

The findings reveal a balanced performance across both macro and micro metrics, indicating that our approach effectively handles bilingual classification tasks. The consistency between these metrics is significant because it shows that the model performs well across all categories, avoiding bias towards any specific class. This is important in multi-label situations, where we want to ensure strong performance across all subject categories in the dataset.

Additionally, the strong scores achieved in both languages highlight the model's multilingual abilities, suggesting that using few-shot classification

[5]https://mlco2.github.io/codecarbon/

and effective label representation allows for accurate classification without requiring extensive fine-tuning. This is especially important in situations with limited resources, where creating large annotated datasets is difficult. The low resource usage highlighted in Table 2, combined with the strong results, shows that employing an efficient framework that includes data augmentation and in-batch contrastive learning can optimize performance even with a minimal dataset. By reducing the amount of data required to achieve high performance, we also decrease the training time and resources needed for the process leading to a significantly more energy-efficient method.

| Metric | value |
|---|---|
| macro-precision | 0.4683 |
| macro-recall | 0.4683 |
| macro-f1 | 0.4683 |
| micro-precision | 0.4725 |
| micro-recall | 0.4725 |
| micro-f1 | 0.4725 |

Table 3: Overall result of the proposed approach: This table presents the evaluation metrics related to the performance of the model. These metrics provide a comprehensive assessment of the model's effectiveness in classifying bilingual technical records, indicating its ability to balance precision and recall across different classes.

| Metric | de | en |
|---|---|---|
| macro-precision | 0.4649 | 0.4745 |
| macro-recall | 0.4649 | 0.4745 |
| macro-f1 | 0.4649 | 0.4745 |
| micro-precision | 0.4690 | 0.4787 |
| micro-recall | 0.4690 | 0.4787 |
| micro-f1 | 0.4690 | 0.4787 |

Table 4: Overall result of the proposed approach by language: This table summarizes the performance metrics for the model's classification of bilingual technical records, presented separately for German (de) and English (en).

## 5 Future Works

While the results demonstrate a solid foundation, there remains a clear opportunity for refinement. Identifying the areas where performance—especially in cases with lower scoring—will be vital for future research.

One challenge we encountered was defining negative pairs for our multi-labeled classification problem. Due to the nature of multi-label classification, we need to manage negative pairs carefully within

a batch, which limits our ability to use techniques like Multiple Negative Ranking Loss (Henderson et al., 2017) that consider all other samples in the batch as negatives. Improving the handling of negative pairs will be important for enhancing the quality of our embeddings in future work.

Another challenge was the equal contribution of each token in sentence embedding and similarity score, which can introduce some noise. On the other hand, extracting important keywords may add additional computational costs. While Fast-Fit's token-level similarity partially addresses this issue by focusing on the most similar token from one sentence to each token in another, we may not capture the importance of all relevant tokens. Considering the top K most similar tokens could help emulate keyword extraction and potentially improve the model's performance.

Another promising direction is the replacement or combination of FastFit's loss function with a more dynamic alternative, such as Adaptive Thresholding Loss (ATL)(Zhou et al., 2020) or its variants. ATL introduces a learnable "threshold class," which allows the model to dynamically adjust the decision boundary for each positive and negative pair. This would enable the model to learn instance-specific thresholds during training, which could lead to better generalization on unseen data.

## 6 Conclusion

In this paper, we explored the capabilities of large language models (LLMs) in the context of automated multi-label domain classification for bilingual technical records. By leveraging the Fast-Fit package and generating synthetic data, we developed a model that effectively classifies records from the TIBKAT dataset.

Our results demonstrate that the proposed approach has the potential for enhanced classification performance across both German and English, highlighting the model's robustness in handling multi-label scenarios. While the findings are promising, they also reveal areas for improvement.

In summary, the results validate the effectiveness of our approach in automated subject classification while also signaling critical areas for further investigation.

## References

Magdalena Biesialska, Katarzyna Biesialska, and Marta R. Costa-jussà. 2020. Continual lifelong learn-

| Record Type | LANG | macro-precision | macro-recall | macro-f1 | micro-precision | micro-recall | micro-f1 |
|---|---|---|---|---|---|---|---|
| Article | de | 0.3333 | 0.3333 | 0.3333 | 0.5000 | 0.5000 | 0.5000 |
|  | en | 0.4460 | 0.4460 | 0.4460 | 0.4539 | 0.4539 | 0.4539 |
| Book | de | 0.4610 | 0.4610 | 0.4610 | 0.4652 | 0.4652 | 0.4652 |
|  | en | 0.4813 | 0.4813 | 0.4813 | 0.4853 | 0.4853 | 0.4853 |
| Conference | de | 0.4516 | 0.4516 | 0.4516 | 0.4542 | 0.4542 | 0.4542 |
|  | en | 0.4806 | 0.4806 | 0.4806 | 0.4853 | 0.4853 | 0.4853 |
| Report | de | 0.4835 | 0.4835 | 0.4835 | 0.4879 | 0.4879 | 0.4879 |
|  | en | 0.3865 | 0.3865 | 0.3865 | 0.3901 | 0.3901 | 0.3901 |
| Thesis | de | 0.4834 | 0.4834 | 0.4834 | 0.4872 | 0.4872 | 0.4872 |
|  | en | 0.4568 | 0.4568 | 0.4568 | 0.4598 | 0.4598 | 0.4598 |

Table 5: This table presents the quantitative results for Subtask 1, organized by record type and language. It includes key performance metrics that reflect the model's classification accuracy and effectiveness for different types of records, enabling a comprehensive analysis of how language and record type influence overall performance.

ing in natural language processing: A survey. In *Proceedings of the 28th International Conference on Computational Linguistics*. International Committee on Computational Linguistics.

D. R. Cox. 1958. The regression analysis of binary sequences. *Journal of the Royal Statistical Society: Series B (Methodological)*, 20(2):215–232.

Robert Dale. 2021. Gpt-3: What's it good for? *Natural Language Engineering*, 27:113–118.

Jennifer D'souza, Sameer Sadruddin, Holger Israel, Mathias Begoin, and Diana Slawig. 2025. SemEval-2025 task 5: LLMs4Subjects - LLM-based automated subject tagging for a national technical library's open-access catalog. In *Proceedings of the 19th International Workshop on Semantic Evaluation (SemEval-2025)*, pages 2570–2583, Vienna, Austria. Association for Computational Linguistics.

Tianyu Gao, Xingcheng Yao, and Danqi Chen. 2022. Simcse: Simple contrastive learning of sentence embeddings. *Preprint*, arXiv:2104.08821.

Nikolaos Gialitsis, Sotiris Kotitsas, and Haris Papageorgiou. 2022. Scinobo: A hierarchical multi-label classifier of scientific publications. In *Companion Proceedings of the Web Conference 2022*, WWW '22, page 800–809. ACM.

Daniel Greenstein. 2000. Digital libraries and their challenges. *Library Trend*, 49.

R. Hadsell, S. Chopra, and Y. LeCun. 2006. Dimensionality reduction by learning an invariant mapping. In *2006 IEEE Computer Society Conference on Computer Vision and Pattern Recognition (CVPR'06)*, volume 2, pages 1735–1742.

Matthew Henderson, Rami Al-Rfou, Brian Strope, Yun hsuan Sung, Laszlo Lukacs, Ruiqi Guo, Sanjiv Kumar, Balint Miklos, and Ray Kurzweil. 2017. Efficient natural language response suggestion for smart reply. *Preprint*, arXiv:1705.00652.

Bharath Kandimalla, Shaurya Rohatgi, Jian Wu, and C Giles. 2020. Large scale subject category classification of scholarly papers with deep attentive neural networks. *Frontiers in research metrics and analytics*.

Yonghao Liu, Renchu Guan, Fausto Giunchiglia, Yanchun Liang, and Xiaoyue Feng. 2021. Deep attention diffusion graph neural networks for text classification. In *Proceedings of the 2021 Conference on Empirical Methods in Natural Language Processing*, pages 8142–8152, Online and Punta Cana, Dominican Republic. Association for Computational Linguistics.

Niklas Muennighoff, Nouamane Tazi, Loïc Magne, and Nils Reimers. 2023. Mteb: Massive text embedding benchmark. *Preprint*, arXiv:2210.07316.

Ghulam Mustafa, Muhammad Usman, Lisu Yu, Muhammad Tanvir Afzal, Muhammad Sulaiman, and Abdul Shahid. 2021. Multi-label classification of research articles using word2vec and identification of similarity threshold. *Scientific Reports*, 11.

Nobal Niraula, Samet Ayhan, Balaguruna Chidambaram, and Daniel Whyatt. 2024. Multi-label classification with generative large language models. In *2024 AIAA DATC/IEEE 43rd Digital Avionics Systems Conference (DASC)*, pages 1–7.

Mobashir Sadat and Cornelia Caragea. 2022. Hierarchical multi-label classification of scientific documents. *Preprint*, arXiv:2211.02810.

Angelo Salatino, Francesco Osborne, Thiviyan Thanapalasingam, and Enrico Motta. 2019. The cso classifier: Ontology-driven detection of research topics in scholarly articles. In *International Conference on Theory and Practice of Digital Libraries*.

Tim Schopf, Alexander Blatzheim, Nektarios Machner, and Florian Matthes. 2024. Efficient few-shot learning for multi-label classification of scientific documents with many classes. *Preprint*, arXiv:2410.05770.

Ken Shoemake. 1985. Animating rotation with quaternion curves. In *Proceedings of the 12th Annual Conference on Computer Graphics and Interactive Techniques*, SIGGRAPH '85, page 245–254, New York, NY, USA. Association for Computing Machinery.

Autumn Toney and James Dunham. 2022. Multi-label classification of scientific research documents across domains and languages. In *Proceedings of the Third Workshop on Scholarly Document Processing*, pages 105–114, Gyeongju, Republic of Korea. Association for Computational Linguistics.

Lewis Tunstall, Nils Reimers, Unso Eun Seo Jo, Luke Bates, Daniel Korat, Moshe Wasserblat, and Oren Pereg. 2022. Efficient few-shot learning without prompts. *Preprint*, arXiv:2209.11055.

Liang Wang, Nan Yang, Xiaolong Huang, Linjun Yang, Rangan Majumder, and Furu Wei. 2024. Multilingual e5 text embeddings: A technical report. *arXiv preprint arXiv:2402.05672*.

Silvan Wehrli, Bert Arnrich, and Christopher Irrgang. 2024. German text embedding clustering benchmark. *Preprint*, arXiv:2401.02709.

Asaf Yehudai and Elron Bendel. 2024. When llms are unfit use fastfit: Fast and effective text classification with many classes. *Preprint*, arXiv:2404.12365.

Wenxuan Zhou, Kevin Huang, Tengyu Ma, and Jing Huang. 2020. Document-level relation extraction with adaptive thresholding and localized context pooling. *Preprint*, arXiv:2010.11304.

## A    Appendix A: Detailed Result

The Germval team evaluated our subject tagging system. Table 5 shows the detailed quantitative results, including Macro and Micro precision, recall, and F1 scores grouped by record type and language, on the TIBKAT dataset.

## B    Appendix B: Prompts to generate domain class description

- English: *Generate a single, comprehensive paragraph that provides a detailed overview of the field of [LABEL] suitable for a classification label. The tone should be informative and encyclopedic.*

- German: *Erstellen Sie einen einzelnen, umfassenden Absatz, der einen detaillierten Überblick über das Gebiet von [LABEL] bietet und für ein Klassifikationslabel geeignet ist. Der Ton sollte informativ und enzyklopädisch sein.*