# OpenReview forum: "Last Minute at the GermEval-2025 LLMs4Subjects Task: Few-Shot Contrastive Learning for Multilingual Multi-Label Classification"
_GSCL.org/KONVENS/2025/Workshop/GermEval — GermEval25 Oral_

### Official Review · Reviewer_VJUw · 2025-08-08
**Innovative Approach with Limited Thematic Alignment: Constructive Review of GermEval 2025 Submission**

**Rating:** 3
**Confidence:** 4

**Review:**

## Overall Impression

To approach the problem with the newly developed FastFit package is clearly an innovative contribution to the field and will help the community to evaluate the potential of this approach.

The main theme for the task was "The Development of Energy- and Compute-Efficient LLM Systems". While the authors provide an interesting solution to subtask 1, it hardly relates to the theme. Neither do the authors supply an example of a less efficient system, nor do they discuss which aspects of their system offer particularly efficient solutions.

## Strength

The related works section highlights very interesting studies and seems well researched.

Applying the FastFit package is a very interesting idea as it has probably not been tested for subject classification before.

The idea to augment the subject labels by LLM-generated descriptive texts is very innovative, too.

The future works section provides a good critical reflection of the work and highlights promising research directions.

## Weaknesses

### Methodology

The usage of the CodeCarbon package is unclear. To showcase the compute or energy efficiency of the system, this would need to be related to some point of comparison.

### Results

The authors do not relate their own results to that of other team(s). So the claim that results are "competitive" is somewhat exaggerated.

## Feedback for improvement

Check the usage of the phrase "the SemEval-Team". This is no longer SemEval but GermEval.

Try to provide better context for table 2. Just inserting the numbers of Table 2 without any explanation and comparison is meaningless. This is not sufficient to relate to the theme of "Energy- and Compute-Efficient LLM Systems". Try to discuss why your system offers an efficient solution, e.g. because the fine-tuning step requires little resources or low inference costs, etc.

The idea of instance-specific thresholding seems promising. You should continue your research in that direction.

Thank you for your contribution to LLMs4Subjects! You should continue your
promising research!

**Summary:**

This article presents a system developed for the shared task LLMs4Subjects at GermEval 2025. In particular, the team addresses subtask 1 of the shared task: multi-domain classification according to the subject classification system of the TIB. To address the problem the authors employ an open source package called FastFit along with an LLM-based data enhancement method, to prepare the data for the usage with FastFit.

---

### Official Review · Reviewer_Fm8g · 2025-08-14
**Clear structure, may need more explanations**

**Rating:** 3
**Confidence:** 4

**Review:**

Overall impression:
Well structured paper, generally clear description of the system, but more details can be added for readers' understanding of the proposed approach.
Provides good perspectives on the task with an energy-efficient approach of fine-tuning a smaller subset of the data, to also deal with the problem of a skewed dataset. The contrastive method may be adapted for usages in other domains.

Strengths:
- Clear structure of the paper.
- The environmental impact of the training process is tracked, which aligns with the shared task's concern on building energy- and compute-efficient systems.
- Extensive suggestions on future work.

Weaknesses:
- Some explanation is lacking on the process and models.
- "few-shot prompting" is mentioned, but not explained in what manner was it done.
- Results section is relatively brief.
- Authors pointed to table 2 for environmental impact, but the table displays only training metrics and not environmental impacts (energy consumed for RAM, GPU etc.) that the package can track.

Feedback for improvement:
- Figure 1 should be cropped so that the selecting frame would not be showing.
- Check if table 2 is mislabelled, or misreferenced.
- If possible, include the links to your system.
- Change "SemEval team" to "GermEval team".
- Question on "we randomly selected a fixed number of 100 samples from each subject category to create a balanced dataset." > 100 samples is a relatively small size given the total number of samples available. Can you explain why did you decided on 100 instead of a bigger size?
- Is it correct that the size of the dataset used for fine-tuning is 100*28*28=78400? 100 samples from each of the 28 class, then 28 class descriptions for each title+abstract?
- Can you explain why did you choose to fine-tune an instruct model (Multilingual-E5 large-instruct) instead of just an embedding model when the generative nature of the instruct model does not seem necessary for the task?
- It is not completely clear how the normalised similarity score for test samples is calculated and what is the selection threshold the label assignment is based on. Can you add more explanation to it?
- Can you provide the length of the class descriptions generated by GPT-4o-mini, and maybe provide some insights on the choice or any effect of the description lengths?

**Summary:**

The paper presented a system developed for Subtask 1, a multi-label classification task. The authors had taken an contrastive learning approach to learn the similarity between the content of the samples and the generated context of the class labels. Few-shot fine-tuning is done by sampling 100 samples from each class instead of using the full dataset.

The system first generates extra contexts for class lables with an LLM (GPT-4o Mini), then use a fine-tuned LLM (Multilingual-E5 instruct model) for few-shot prompting. Similarity scores between the concatenated title and abstract of each sample and each label description is calculated.

The system emphasised on being energy- and compute-efficient, stating that environmental impacts from training is tracked, but unfortunately did not disclose the values from the tracking.

---

### Decision · Program_Chairs · 2025-08-14

Accept (Oral)